# Seismic Performance of a Single-Story Timber-Framed Masonry Structure Strengthened with Fiber-Reinforced Cement Mortar

**DOI:** 10.3390/ma17153644

**Published:** 2024-07-24

**Authors:** Wei Tan, Tiegang Zhou, Lixin Zhu, Xiang Zhao, Wen Yu, Liangyi Zhang, Zengfei Liang

**Affiliations:** 1School of Civil Engineering, Xi’an University of Architecture and Technology, Xi’an 710055, China; tanwei@xauat.edu.cn (W.T.); zhaoxiang@xauat.edu.cn (X.Z.); 15517806203@163.com (L.Z.); m15529556230@163.com (Z.L.); 2China Academy of Building Research, Beijing 100013, China; 15829735823@163.com (L.Z.); wote@aliyun.com (W.Y.)

**Keywords:** timber-framed masonry structure, fiber-reinforced cement mortar, scaled model, seismic performance, shaking table test

## Abstract

Timber-framed masonry structures are widely used around the world, and their seismic performance is generally poor. Most of them have not been seismically strengthened. In areas with high seismic fortification intensity, there are great potential safety hazards. And it is urgent to carry out effective seismic reinforcement. However, due to the complicated construction process of the existing reinforcement technology, the poor durability of the reinforcement materials, and the significant disturbance to the life of the original residents, an efficient single-story timber-framed masonry structure reinforcement technology suitable for comprehensive promotion and application has not been explored. In this paper, a fiber-reinforced cement mortar (FRCM) material was proposed. A 1/2 scale model of a single-story timber-framed masonry structure was taken as the research object. The method of strengthening a single-story timber-framed masonry structure with FRCM layer was adopted. And the shaking table test of the model before and after reinforcement was carried out in turn. The dynamic characteristics, failure modes, acceleration response and displacement response of the FRCM layer-strengthened structure were analyzed through comparisons of the two cases. The experimental results showed that the FRCM layer significantly improved the seismic performance of the seismic-damaged single-story timber-framed masonry structures. The X- and Y-direction natural frequencies of the model structure were increased by 31.30% and 30.22%, respectively, after the structure was strengthened with FRCM. During a rare eight-degree earthquake, the inter-story displacement angles in the X- and Y-direction of the unreinforced model reached 1/98 and 1/577, respectively, and the structure was destroyed, while the inter-story displacement angle of the FRCM-reinforced model was only 1/2 of that the unreinforced model. During a rare nine-degree earthquake, the X-direction inter-story displacement angle of the model strengthened with FRCM reached 1/78 and the Y-direction inter-story displacement angle reached 1/178. At this time, the reinforced model structure was destroyed, but there was no collapse of the structural components, which met the seismic design objectives of “operational under the design minor seismic intensity, repairable damage under the design seismic precautionary intensity, and collapse prevention under the design rare seismic intensity”, which proved that the FRCM layer was an effective and feasible way to strengthen the existing single-story wood-masonry rural building.

## 1. Introduction

Masonry structure is one of the most widely used structural forms of existing rural buildings in the world, and it is mainly distributed in vast rural areas [1,2,3]. Rural buildings with masonry structures have been seriously damaged in previous earthquakes in recent years, showing obvious brittle failure characteristics [4,5,6,7,8].

At present, there are still many single-story timber-framed masonry structures in the area where the seismic fortification intensity is eight degrees, and most of the houses were built earlier. Most of them adopted the mixed bearing system of brick wall and roof woodwork. The main features were that the rear longitudinal wall and the pediment wall on both sides did not have open doors and windows, the front facade had no load-bearing wall, doors and windows were arranged longitudinally, and only the cantilevered wall was set [9,10,11], as shown in Figure 1. The masonry quality of the wall was poor, and there were no ring beams and structural columns, which were in the “unfortified” state, and there were serious safety hazards. Therefore, it was of great significance to adopt a reinforcement method with economical cost, convenient operation, and remarkable seismic effect [12,13].

In recent years, masonry structure has attracted the attention of scholars from various countries, and a large number of experimental studies have been carried out [14,15,16,17,18]. A variety of commonly used reinforcement methods for masonry structure houses have been summarized and proposed [19,20,21], and the effectiveness of the reinforcement methods has been verified, but they also show different limitations. Based on a quasi-static test on a low-strength brick wall reinforced by a single-sided cement mortar surface layer, it was found that the reinforcement method has a better reinforcement effect on the worse seismic performance of the original wall, but the overall shear bearing capacity is limited [22]. Through experimental research, Tang Caoming found that the seismic performance of brick walls reinforced with a single-sided steel mesh cement mortar surface layer was not significantly better than that of brick walls reinforced with single-sided cement mortar surface layer. When the vertical compressive stress was increased, the seismic performance significantly improved [22]. Deng Mingke used high ductile concrete to reinforce the masonry structure, which enhanced the lateral stiffness and bearing capacity of the masonry structure and enhanced the seismic performance [23,24,25]. However, high ductility concrete is a new type of building material, which is mainly used in cities and surrounding villages and towns. The application of high-ductility concrete in remote villages and towns is less economical and the construction cost is higher [26,27]. Tan tested FRP-reinforced brick walls and found that compared with unreinforced walls, the in-plane and out-of-plane strength of the wall were significantly improved [28,29]. However, in practical engineering applications, these walls also have problems such as poor durability of adhesive materials and inconvenient construction in humid and low-temperature environments [30].

Summarizing the previous research results of scholars, it was found that the current reinforcement materials and reinforcement technology for single-story timber-framed masonry structures cannot be widely promoted and applied due to poor material durability, complex reinforcement methods, the cumbersome reinforcement process, poor economy, and the significant disturbance to the lives of the original residents. In this paper, a new type of FRCM material is proposed. It is verified by a shaking table test that the material has a significant effect on the seismic performance of single-story timber-framed masonry structures. This study provides a new type of reinforcement material and reinforcement method for the seismic reinforcement of single-story timber-framed masonry structures, which has the value of popularization and application.

## 2. Experiment Program

In order to improve the seismic performance of single-story timber-framed masonry structures, a new type of FRCM material was proposed in this paper, and the mechanical properties of the reinforcement material were tested. In order to verify the effectiveness of the reinforcement material, a 1/2 scale model was designed and manufactured based on a three-bay single-story timber-framed masonry structure in the eight-degree fortification area, with the site categorized as Class II and the design group as the first group [31]. The mechanical properties of the materials used in the model were analyzed. The shaking table test of the model specimens before and after the FRCM-layer reinforcement was carried out, which provides data support for the subsequent analysis of the seismic performance indexes such as dynamic characteristics, acceleration response, and displacement response before and after the failure of the model specimens.

### 2.1. The Design and Fabrication of Specimens

Considering the limitations of the size and bearing capacity of the shaking table, a 1/2 scale model is designed and manufactured. The wall is made of MU10 bricks with a size of 115 mm × 53 mm × 30 mm, the thickness of the wall is 120 mm, and the strength grade of the masonry mortar is M1. The less artificial mass model is adopted. The purlin is paved with 10 mm thick wood board and 100 mm thick grass mud is added to the board. The mass of the model was 6.41 tons after adding the weight. The layout and dimensions of the model structure are shown in Figure 2.

The test model structure was fabricated at the Integrated Laboratory of Earthquake Engineering of HUIXIAN. The fabrication process is shown in Figure 3. According to the requirements of relevant standards [32], the test brick is soaked one day in advance, the moisture content is controlled at about 65%, and the ambient temperature is about 20 degrees. The wall is built by an intermediately skilled worker using the same masonry method as the prototype house, and the thickness of the mortar joint is 5 mm. Due to the limitations of the experimental conditions, the model structure was subjected to ground motion loading after 3 months. Relevant research shows that the strength growth rate of masonry after 28 days of curing is low, so considering the curing time has little effect on the test results [33]. After the model was destroyed, the FRCM layer was used for reinforcement. After 4 weeks of maintenance, the ground motion loading was carried out again until the model was destroyed.

The plane layout of the prototype structure is irregular, the opening area of the front longitudinal wall is too large, and the gables on both sides lack effective pulling measures. The out-of-plane stiffness of the gable is low and the stability is poor. The double side-layer reinforcement can effectively improve the wall stiffness, and the full-length screw can effectively enhance the pulling between the gables. In addition, through other scholars’ research, it is found that the reinforcement method involving a cement mortar reinforcement strip can also improve the bearing capacity, degeneration ability and seismic capacity of masonry wall [34,35]. Combined with the actual earthquake disaster and the failure phenomenon of unreinforced model, the reinforcement scheme of this study is formulated.

Reinforcement mainly includes wall crack repair and main structure reinforcement. The operation steps are as follows: The cracks around the wall are polished and the glue injection bases are pasted at a wide crack. The walls are reinforced with FRCM and maintained for 7 days. The pressure syringe is used to fill the sealing glue from bottom to top until the uppermost base has the sealing glue flowing out and the base is closed. A long screw is set along the eaves of the front longitudinal wall to pull the gables on both sides. Square wood scissors are used to strengthen the connection between wooden roof trusses.

Reinforcement with FRCM layer, the inside and outside of the gables was plastered to increase the stiffness and integrity. The rear longitudinal wall was reinforced with strips, the width of the vertical reinforcement strips on the outer side of the wall was 500 mm, the width of the vertical reinforcement strips in the middle was 400 mm, and the width of the horizontal reinforcement strips was 300 mm. The width of the vertical reinforcement strip set at the inner corner of the rear longitudinal wall was 200 mm. The thickness of FRCM layer was 10 mm, as shown in Figure 4. The reinforcement process and effect of the model are shown in Figure 5.

### 2.2. Material Properties

#### 2.2.1. Bricks

The strength grade of the brick used in the prototype structure is MU10, and the model structure adopts the same grade of brick. Experimental bricks were taken from a building materials market in Beijing. According to the relevant standard requirements [36], 10 brick samples are selected, the loading rate is 5 kN/s, and the compressive strength is 11.2 MPa. The elastic modulus is 7652 MPa. The water absorption rate of the sample soaked in water at room temperature for 24 h is 13.8%.

#### 2.2.2. Masonry Mortar

The strength grade of masonry mortar of prototype structure is M1. According to the requirements of the technical standard [37], the model structure masonry mortar was prepared according to m_cement_:m_sand_:m_water_ = 1:6.83:1.61, and six groups of 18 cubes with a size of 70.7 mm × 70.7 mm × 70.7 mm were formed. Cement and sand were taken from a building materials market in Beijing. The compressive strength test was carried out at a loading rate of 0.5 kN/s. The compressive strength of the mortar test block cured for 28 days was measured 1.4 MPa.

#### 2.2.3. Wood

According to the requirements of relevant standards [38], the same pinewood of the prototype structure was selected for the test. The pinewood was taken from a timber market in Beijing. The measured wood moisture content is 13.8%, the density is 510 kg/m^3^, the compressive strength parallel to the grain is 13.5 MPa, the elastic modulus is 10,250 MPa, and the Poisson’s ratio is 0.4.

#### 2.2.4. FRCM

The FRCM material used in this paper is made of cement, fly ash, slag, sand, water reducing agent and fiber. The materials are all from a chemical market and building materials market in Beijing. The mixing ratio is 1:0.5:0.5:4:0.02:0.006, and the water–binder ratio is 0.3. According to the relevant requirements of Literature [33], the powdered material is first mixed and stirred evenly and then the fiber is evenly mixed into the mixture several times, and then the mixture of water reducing agent and water is sprayed evenly one by one, stirring for 4 min to prepare fiber-reinforced cement mortar material, as shown in Figure 6. According to the requirements of the relevant technical standards [39,40], three test blocks with a size of 100 mm × 100 mm × 100 mm were created for the compressive strength test. The loading rate was 5 kN/s and the compressive strength of 28d was measured to be 40 MPa. The specimen size of the flexural test and the equivalent bending test is 40 mm × 40 mm × 160 mm, and the loading rate is 0.05 kN/s. The flexural strength at 28d age is 16.2 MPa, the ratio of flexural strength to compressive strength is 0.405, and the equivalent flexural strength at 28d age is 5.15 MPa. The loading equipment is WE-30 universal material testing machine.

The failure phenomenon and load–deflection curve are shown in Figure 7. It can be seen from the curve that the equivalent bending test of the FRCM is divided into the elastic stage, plastic cracking stage, and failure stage. The deflection increases with the increase in the load. After the specimen cracks, the deflection continues to increase and the load decreases rapidly. At this time, the fibers randomly distributed inside the specimen begin to be pulled, and the load begins to increase slowly until it reaches the maximum load.

### 2.3. Similarity Relation

The model structure with a scale ratio of 1/2 was used in the test. The similarity relationship of the model structure is shown in Table 1.

### 2.4. Instrumentation and Measurement

During the shaking table test, the X-direction was defined as the longitudinal direction (east–west direction) of the model and the Y-direction as the transverse direction (north–south direction) of the model. According to the damage severity and structural response of different positions of the structure in the actual earthquake disaster, acceleration sensors and displacement sensors were mainly arranged in the foundation beam, at the height of the wall eaves and at the height of the roof [41,42]; the specific location is as follows.

A total of nine acceleration sensors were arranged, including one on the shaking table, one on the model foundation, four in the middle of the height of the wall eaves, two on the top of the pediment wall of the east and west gables, and one in the middle of the ridge purlin. The location and number are shown in Figure 8.

A total of 6 displacement sensors were arranged along the X- and Y-direction at the southwest corner position of the foundation beam, the middle of the eaves of the western gable wall, and the top of the pediment of the western gable wall. The location and number are shown in Figure 9.

### 2.5. Acceleration Time History Curves

The prototype structure is located in the eight-degree fortification area. The characteristic period is 0.35 s and the horizontal seismic influence coefficient is 0.45. Three seismic waves are selected. The EL Mayor–Cucapah wave (hereinafter referred to as the EL Mayor wave) is an earthquake record that occurred in Mexico in 2010, with a moment magnitude of 7.2 and a ground motion duration of 100 s. The Landers wave is a record of the Landers earthquake that occurred in the United States in 1992, with a moment magnitude of 7.28 and a ground motion duration of 50 s. The artificial wave is fitted by seismic fortification intensity, site category, a characteristic period, etc., and the ground motion duration is 20 s. The acceleration time history curves are shown in Figure 10.

### 2.6. Test Program

The shaking table test was carried out in the Integrated Laboratory of Earthquake Engineering of HUIXIAN, Institute of Engineering Mechanics, China Earthquake Administration. The shaking table equipment is developed by Beijing Baoke Test System Limited Company (Beijing, China). The table size is 5m × 5m, and the maximum bearing capacity is 30 t. Before the test loading and after each input ground motion, the white noise with a peak acceleration of 0.05 g was used to sweep the frequency. El Mayor, Landers and artificial waves were input step by step to analyze the failure characteristics, natural vibration period, acceleration response, and displacement response of the model structure. The unreinforced model structure (M1) was loaded from the 1st to the 17th loading conditions, and the reinforced model structure (M2) was loaded from the 1st to the 21st loading conditions, as shown in Table 2.

## 3. Test Result Analysis

### 3.1. Experimental Phenomenon

#### 3.1.1. Experimental Phenomenon of M1

No noticeable cracks were detected on M1 after the first stage of loading (PGA = 0.13 g), and only slight shaking was observed. During the second stage of loading (PGA = 0.37 g), the roof truss swayed slightly along the longitudinal direction, accompanied by friction noise, and vertical cracks appeared between the cantilevered wall and the wooden column.

During the third stage of loading (PGA = 0.74 g), the model exhibited a violent shaking phenomenon, accompanied by obvious sound. In the 14th loading condition, two cracks were found at the top height of the ridge wall in the east and west gables, accompanied by obvious tension and extension. After the 16th loading condition, the cracks on the gable continued to extend to form through cracks. The angle between the upper cracks and the horizontal direction was about 30°. The lower cracks of the east gable were about 15° downward. The lower cracks of the west gable were approximately horizontal cracks, and the flashing occurred for the gable at the crack position. At this time, it was judged that the model structure had been seriously damaged and the loading was stopped, as shown in Figure 11.

#### 3.1.2. Experimental Phenomenon of M2

No noticeable cracks were detected on M2 after the first stage of loading and the second stage of loading (PGA = 0.13 g, PGA = 0.37 g).

During the third stage of loading (PGA = 0.74 g), cracks began to appear in M2. Under the 14th loading condition, the nourishing sound of the fiber being pulled off could be heard, but there was no crack in the model structure. After the loading of the 16th loading condition, a crack with a length of 100 mm appeared on the top of the west side of the rear longitudinal wall. Under the 18th loading condition, vertical cracks appeared on the east side of the rear longitudinal wall, and the vertical cracks on the west side developed downward. The east and west cantilevered walls had a vertical seam with a width of 1 mm along the vertical gray seam at a distance of 100 mm from the gable. The lower right corner of the east gable produced cracks with a width of 1 mm and a length of about 300 mm from bottom to top. There was no obvious damage to the main structure of the model, and it could still be loaded.

During the fourth stage of loading (PGA = 1.15 g, the 20th loading condition), the cracks in the rear longitudinal wall were obviously opened and closed, and the surrounding mortar dropped. The cracks continued to extend downward to form a through crack with a width of 5 mm. The rear longitudinal wall was divided into three parts by the cracks, as shown in Figure 8. There was a 45-degree oblique crack inside the gable on the east side, which continued to extend upward to the purlin to form a joint with a width of 5 mm. The outer drum phenomenon appeared on the outer eaves of the wall, as shown in Figure 12. The bricks in the middle and lower parts of the wall were misplaced, and the bricks fell off the top and middle parts. The wood column was separated from the wall, and the bricks and mortar in the west corner were crushed and dropped. The FRCM layer peeled off and the corner position was seriously damaged, as shown in Figure 13. At this time, it was considered that the model structure had been destroyed and loading was stopped, but the structure and the main force components had not collapsed, as shown in Figure 14.

#### 3.1.3. Failure Mechanism

According to the M1 failure mode, the damage of the model structure mainly occurs on both sides of the gables. The main reason is that the gables on both sides lack out-of-plane support and connection. In addition, due to the low strength of the masonry mortar of the model structure wall and the poor masonry technology of the wall, the integrity of the gables on both sides is worse, so the damage mainly occurs on both sides of the gables. After the FRCM surface layer reinforcement, the integrity of the gables on both sides of M2 greatly improved. In addition, the longitudinal through-length screw set up at the eaves effectively pulled the gables. Therefore, the damage of M2 mainly occurred in the weak part of the FRCM strip reinforcement of the rear longitudinal wall, and the longitudinal wall does not show the phenomenon of external flash. There is less gable damage than was observed for M1, which verifies the effectiveness of the reinforcement.

### 3.2. Dynamic Characteristics Analysis

Before and after the loading of each level of seismic action, the model structure is scanned by white noise, and the natural frequency and damping ratio of the model structure after different levels of seismic action are analyzed, as shown in Table 3.

According to the changes in the natural frequency and damping ratio of the model under different loading conditions, the damage degree of the model structure can be evaluated. The following conclusions can be drawn from Table 3:

(1) After the loading of M1, the natural frequencies in X- and Y-directions were measured to be 14.160 Hz and 12.890 Hz, respectively. After the reinforcement of FRCM layer, the initial frequencies of M2 were 18.592 Hz and 16.786 Hz, and the natural frequencies in X and Y directions were increased by 31.30% and 30.22%. This shows that the reinforcement with FRCM layer had a significant effect on the overall stiffness of the model structure.

(2) After the M1 was loaded under the action of an eight-degree earthquake of Landers wave, the white noise scanning showed that the X- and Y-direction frequencies were reduced by 18.1% and 17.0%, respectively, compared with the initial frequency, and the natural frequency was greatly reduced. When M2 was loaded to this condition, the X- and Y-direction frequencies decreased by 10.5% and 11.1%, respectively, compared with the initial frequency, and the decrease in the natural frequency was significantly smaller. This shows the effectiveness of FRCM layer reinforcement in improving structural stiffness. Under this condition, the X- and Y-direction damping ratios of M1 increased by 49.9% and 104.6%, respectively, and that of M2 increased by 70.9% and 127.1%, respectively. The main reason is that M2 is reinforced by the FRCM surface layer on the basis of M1 earthquake damage. The interior of the wall has been damaged, and there is friction during the vibration process, so the damping ratio is large. When loaded to the El Mayor wave nine-degree earthquake, the M2 X- and Y-direction frequencies decreased by 25.1% and 25.5% from the initial frequency, and the natural frequency was greatly reduced, but the model’s structure remained intact and no collapse occurred. The main reason is that the FRCM layer not only improves the overall stiffness of the model structure but also greatly improves the integrity and stability of the wall.

(3) The X-direction natural frequency of M1 began to decrease significantly at the 15th loading condition, indicating that obvious damage occurred after the 14th loading condition. In the 11th loading condition, the measured Y-direction frequency of the M2 was significantly reduced, indicating that the M2 was obviously damaged after the 10th loading condition.

(4) Under the third loading condition, the natural frequency of the M2 was measured and was found to be larger. The main reason is that the M1 is reinforced with the FRCM layer immediately after the end of loading, and the model structure wall is in the overall unstable state of damage. Under the action of loading condition 2, the vibration deformation of the wall is restored, making it stable, and the stiffness of the wall is slightly increased, so the natural frequency of the model is slightly larger. With the input of seismic action under different loading conditions, the natural frequencies of model 1 and model 2 continue to decrease, the damage degree of model structure increases cumulatively, and the stiffness of model continues to decrease, as shown in Figure 15.

It can be seen from Figure 11 that the damping ratio of the model showed an increasing trend. Because the FRCM layer of the rear longitudinal wall was reinforced by strips, fine cracks were found at the weak position of the rear longitudinal wall reinforcement with the step-by-step input of seismic action. When the model was loaded to the 20th loading condition, vertical penetrating cracks appeared in the wall, and the width surpassed 5 mm. The rear longitudinal wall was divided into three sections by the vertical cracks, but the roof structure remained whole, which played a good role in restraining the wall. The relative motion between the walls of the rear longitudinal wall consumed a lot of seismic energy. In addition, the slip dislocation between the wood members and the wall further consumed part of the energy, the structural energy consumption increased, and the damping ratio increased. The above analysis showed that the overall stiffness of the structure could be effectively enhanced by using the FRCM layer to reinforce the earthquake-damaged timber-framed masonry structure. During the seismic action, the stiffness degradation was relatively slow, and the damage resistance and seismic capacity of the structure were enhanced.

### 3.3. Acceleration Response Analysis

The acceleration amplification factor is generally determined by the ratio of the maximum measured acceleration at the measuring point of the model structure to the maximum input acceleration of the platform. Additionally, it can directly reflect the dynamic response of the model structure and indirectly reflect the seismic damage degree of the model structure. The acceleration amplification coefficients at different positions of the model structure in the X- and Y-directions are shown in Table 4 and Table 5.

From Table 4 and Table 5, it is found that with the continuous loading, the acceleration amplification coefficients of M1 and M2 are fluctuating, mainly due to the different predominant frequencies of each seismic wave, which causes the acceleration response of the model structure to be different. When the natural frequency of the model structure is within the predominant frequency range of a certain seismic wave, the seismic response of the model structure is relatively strong. The model structure will also be damaged, resulting in a decrease in stiffness, and the natural frequency will change accordingly. It is possible to be within the predominant frequency range of other seismic waves, resulting in a strong seismic response [43].

Due to the slight damage inside the model structure after loading, inelastic damage gradually develops, the structural stiffness changes, the coherence of the structure in the height direction decreases, and the acceleration transmission capacity decreases. Therefore, with the increase in the input seismic action, the acceleration amplification factor shows a decreasing trend as a whole, as shown in Figure 16, Figure 17 and Figure 18.

(1) Under the action of the same seismic wave, with the increase in the input acceleration of the table, the X- and Y-direction acceleration peaks at the roof ridge and eaves of M1 and M2 gradually increase, but the increase gradually decreases. The main reason for this is that with the increase in the input acceleration, the internal damage of the model gradually accumulates, the stiffness decreases, and the acceleration amplification coefficient decreases.

(2) Before the 14th loading condition, with the increase in height, the acceleration of M1 shows an increasing trend as a whole. The main reason is that before this condition, the model structure is not obviously damaged; the integrity of the roof structure, the integrity of the roof structure and the wall, and the integrity of the wall itself remain good; the stiffness change is small; the model natural frequency decreases slightly; and the acceleration amplification coefficient at the ridge is large. After the 14th loading condition, with the increase in height, the acceleration of M1 increases first and then decreases. The main reason is that when loading, horizontal cracks occur on both sides of the gable, the stiffness of the gable decreases, the natural frequency of the model decreases obviously, the damping ratio increases, and the acceleration amplification coefficient at the height of the eaves and the height of the roof ridge decreases obviously. The friction dislocation occurs between the roof structure and the rear longitudinal wall, and the acceleration amplification coefficient at the roof ridge decreases more obviously.

(3) Before the 10th loading condition, with the increase in height, the acceleration of M2 shows an increasing trend as a whole. The main reason is that before the loading of M2, square wood scissors are used to connect the roof trusses, the wall cracks are repaired, and the wall is reinforced by the FRCM layer. The overall stiffness of the model has been strengthened to a certain extent. Before this condition, the damage to the model is small, the stiffness changes little, and the acceleration amplification coefficient at the roof ridge is large. After the 10th loading condition, there is a dislocation between the roof structure and the wall, the integrity is weakened, the damping ratio is increased, and the acceleration amplification coefficient at the roof ridge is greatly reduced. However, at this time, there is basically no damage to the wall, the stiffness of the wall is reduced by a small margin, and the acceleration amplification factor at the eave height is reduced by a small margin.

In addition, the acceleration amplification factor at the roof ridge and eaves is greatly reduced under the 20th loading condition. Combined with the analysis of the experimental phenomena during loading, the model structure is seriously damaged, and the stiffness of the model structure is significantly reduced.

### 3.4. Displacement Response Analysis

The displacement curves of different positions of the model can be obtained by the displacement sensors arranged on the model, and the displacement response values of the bottom beam, the middle position of the gable height of the west hill wall, and the top of the peak wall of the west hill wall are mainly collected in this test. Interlayer displacement generally refers to the difference in horizontal displacement between adjacent floors above and below the house, and the interlayer displacement in this test refers to the displacement at the top of the peak wall minus the displacement at the height of the bottom beam, as shown in Table 6.

(1) After the loading of the 12th loading condition, the top of the south side of the gable wall of M1 began to exhibit fine cracks. When loaded to the 14th loading condition, the interlayer displacement of M1 along the X direction increased sharply. When the loading of the 16th loading condition was completed, the maximum interlayer displacement in the X direction reached 20.983 mm and the interlayer displacement angle was 1/98. At this time, the gable wall had obvious external flashing, and the wall had horizontal through cracks and oblique 30-degree through-length cracks, and the model structure was seriously damaged. The main reason for this is that the quality of the wall of the model structure is poor, the integrity of the roof is weak, the connection between the gables is poor, and the overall stiffness of the model structure is poor, resulting in large deformation.

(2) Under the same loading conditions, the interlayer displacements of M2 are smaller than those of M1 in both the X- and Y-direction. The main reason is that the M1 structure is not reinforced, and the integrity of the wall and the out-of-plane stability of the gable are poor. After the reinforcement of the FRCM surface layer, the integrity of the wall improves, and the out-of-plane stiffness of the gable also increases. In addition, the installation of a long screw also limits the displacement of the gable.

(3) When loading to the level of an eight-degree rare earthquake, the interlayer displacement of M1 increases sharply, the interlayer displacement angle in X direction increases from 1/987 to 1/186, and the interlayer displacement of M2 is only 1/579, which indicates that the FRCM layer reinforcement can effectively improve the stiffness and deformation capacity of single-story timber-framed masonry structure.

(4) Both M1 and M2 show that the interlayer displacement in X direction is greater than that in Y direction. The FRCM layer provides double-sided reinforcement on the gable wall, which increases the in-plane stiffness and also increases the out-of-plane stiffness. Due to the insufficient constraint of the front longitudinal wall on the gable wall, the out-of-plane stiffness (X-direction) of the gable wall is smaller than the in-plane stiffness (Y-direction), and therefore the displacement in X direction is larger.

## 4. Conclusions

In this paper, a new type of FRCM material was proposed, and the mechanical properties of the reinforcement material were tested. A 1/2 scale model was designed and manufactured, and the mechanical properties of the materials used in the model were tested and analyzed. Shaking table tests were carried out on the model specimens before and after the reinforcement of the FRCM layer. The dynamic characteristics, failure modes, acceleration response and displacement response of the two model structures under different seismic actions were compared and analyzed. The effectiveness of the layer reinforcement with FRCM material was verified, providing a new type of reinforcement material and reinforcement method for the seismic reinforcement of single-story timber-framed masonry structures, and this method has the value of popularization and a range of applications. The main conclusions are as follows:Under the action of an eight-degree rare earthquake of Landers wave, M1 is seriously damaged and the overall performance is poor. When M2 is loaded to the level of a nine-degree rare earthquake, it is damaged, but there is no collapse of the structure or its components, which maintain good integrity. The integrity of the model structure is enhanced by the reinforcement of the FRCM layer.Under the same intensity of earthquake, the interlayer displacement of M2 in X- and Y-directions is significantly smaller than that of M1. When M1 is destroyed, the inter-layer displacement angles in X- and Y-directions reach 1/98 and 1/577, respectively, and the inter-layer displacement angle of M2 is only 1/2 that of M1. The deformation capacity of the model structure is significantly improved by the reinforcement of the FRCM layer.After loading, the natural frequencies of M1 in X- and Y-directions are 14.160 Hz and 12.890 Hz, respectively. The initial natural frequencies of M2 are 18.592 Hz and 16.786 Hz. The natural frequencies of X- and Y-directions are increased by 31.30% and 30.22%, respectively. The stiffness and seismic performance of the model structure can be greatly improved by the reinforcement of the FRCM layer.After the earthquake action, the cracks of M1 mainly appear on the load-bearing gables on both sides, while the cracks of M2 are mainly concentrated on the rear longitudinal wall, and there are no obvious stress cracks on the load-bearing gables on both sides, indicating the effectiveness of the reinforcement method. However, the front eaves of the eastern gable are slightly flashed. The main consideration is that the connection between the gables is slightly insufficient, and further follow-up research should be considered to strengthen the treatment. This reinforcement method is only suitable for the reinforcement of a single-story timber-framed masonry structure, and our research group is currently working on the reinforcement of multi-story timber-framed masonry structures.

The use of an FRCM layer to reinforce an earthquake-damaged single-story timber-framed masonry structure significantly enhanced the stiffness of the gable wall and improved the structural integrity. It therefore meets the seismic design objectives of “operational under the design minor seismic intensity, repairable damage under the design seismic precautionary intensity, and collapse prevention under the design rare seismic intensity”. The FRCM layer has a good effect on strengthening single-story timber-framed masonry structures and has promotion value.

## Figures and Tables

**Figure 1 materials-17-03644-f001:**
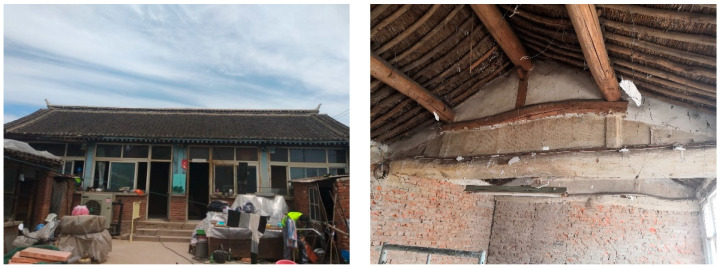
Timber-framed masonry structure.

**Figure 2 materials-17-03644-f002:**
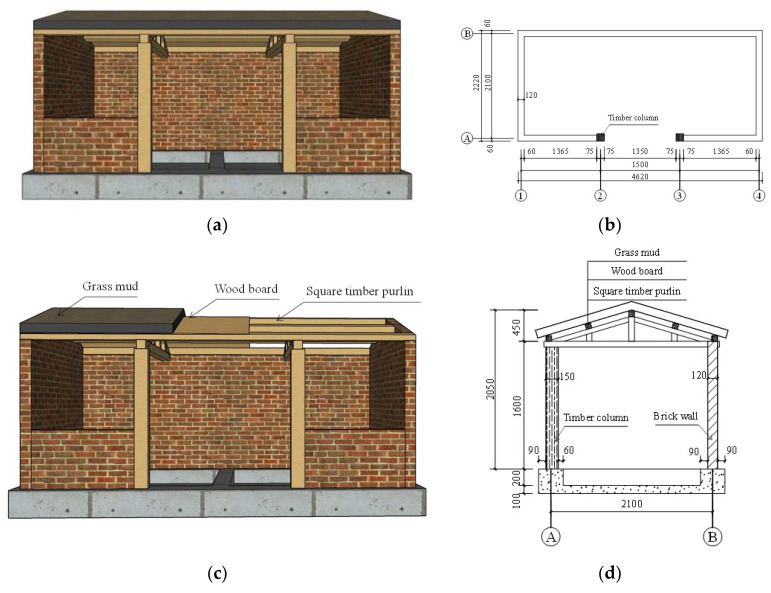
Layout and dimensions of the model structure (mm): (**a**) Front façade; (**b**) layout plan; (**c**) roof structure; (**d**) structural details.

**Figure 3 materials-17-03644-f003:**
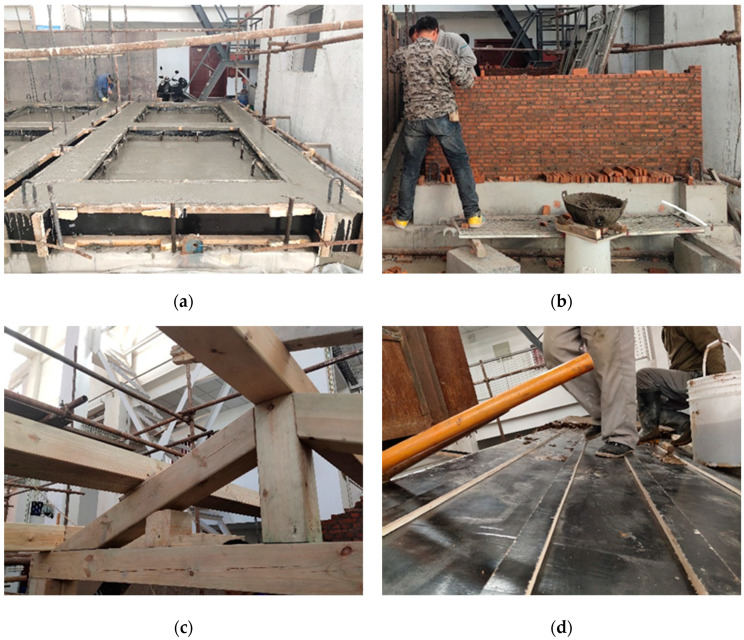
The fabrication process of model structure: (**a**) Pouring of foundation beam; (**b**) masonry and building of the wall; (**c**) installation of roof woodwork; (**d**) construction of the roof.

**Figure 4 materials-17-03644-f004:**
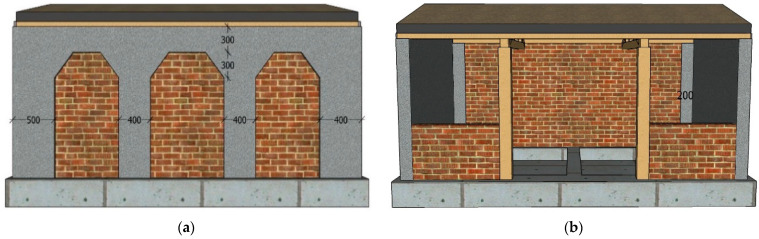
Model structure reinforced with FRCM layer (mm): (**a**) Reinforcement of the outside of the rear longitudinal wall; (**b**) reinforcement of the inside of the rear longitudinal wall and the gable.

**Figure 5 materials-17-03644-f005:**
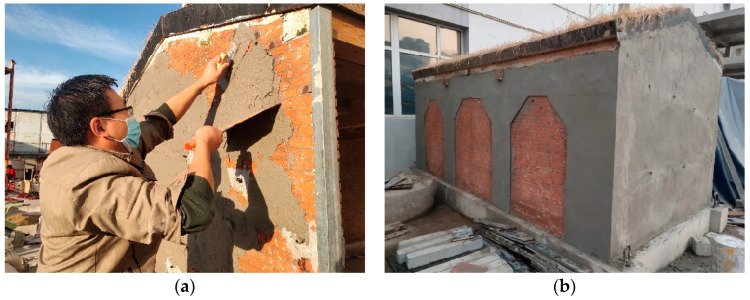
Model reinforcement process and effect: (**a**) Model reinforcement process; (**b**) model reinforcement effect.

**Figure 6 materials-17-03644-f006:**
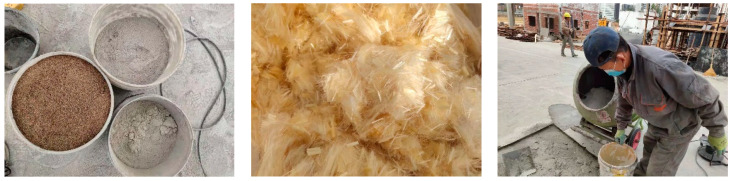
FRCM materials and mixing process.

**Figure 7 materials-17-03644-f007:**
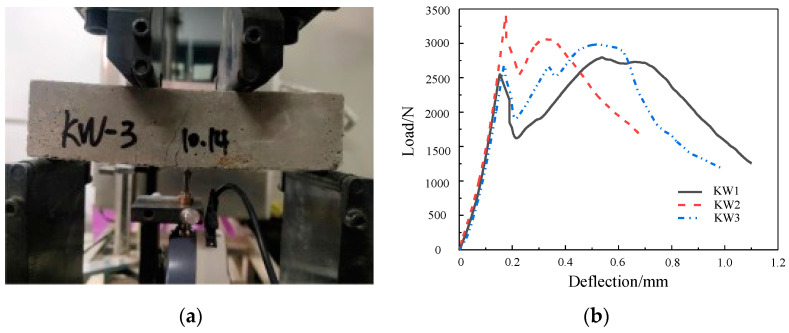
Equivalent bending test and load–deflection curve: (**a**) Failure phenomenon; (**b**) load–deflection curve.

**Figure 8 materials-17-03644-f008:**
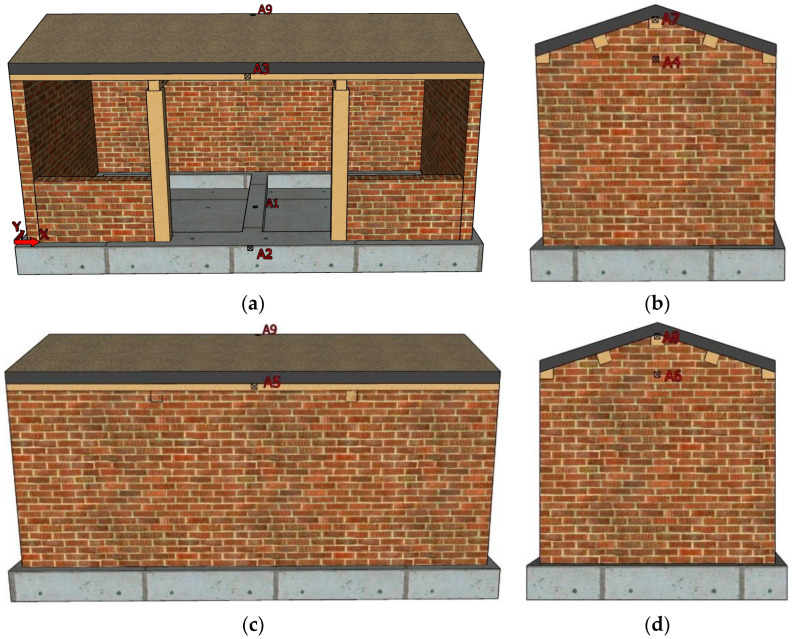
Layout of accelerometers: (**a**) South elevation; (**b**) east elevation; (**c**) north elevation; (**d**) west elevation.

**Figure 9 materials-17-03644-f009:**
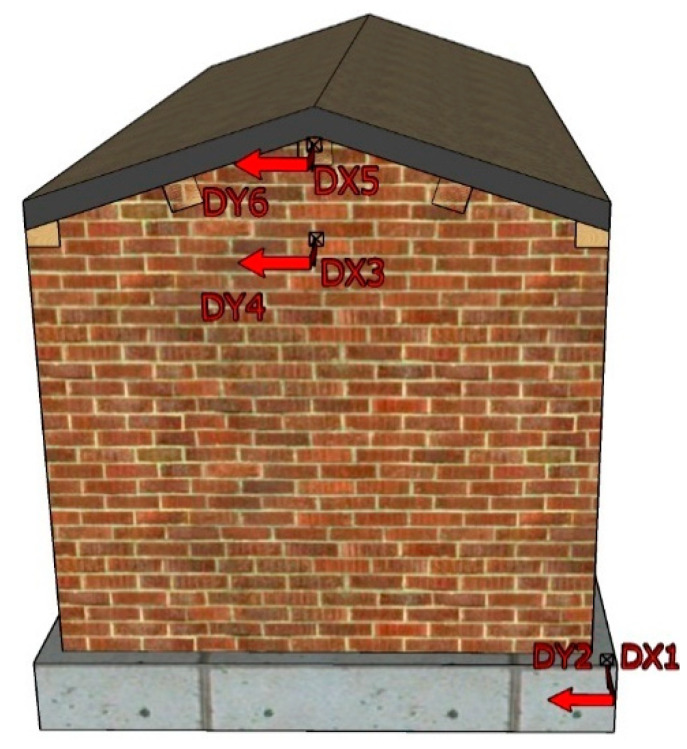
Layout of displacement meters.

**Figure 10 materials-17-03644-f010:**
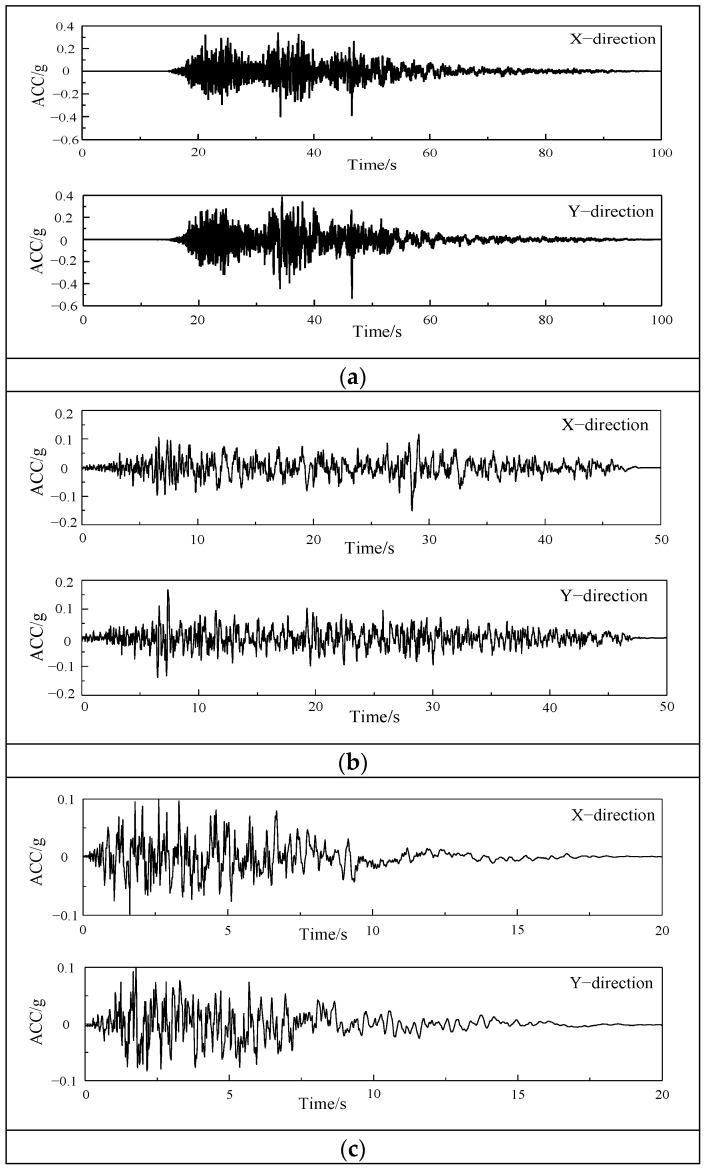
Acceleration time history curves: (**a**) El Mayor wave; (**b**) Landers wave; (**c**) artificial wave.

**Figure 11 materials-17-03644-f011:**
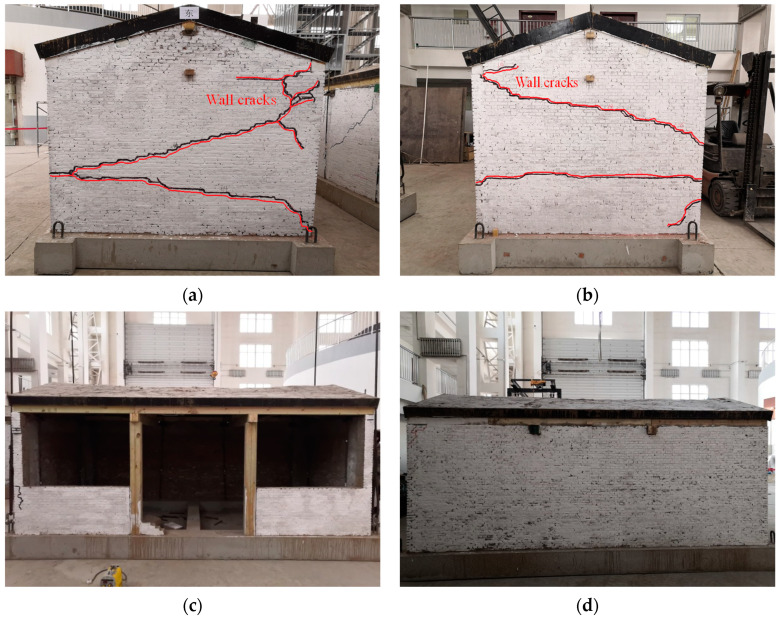
Destruction mode of the model structure: (**a**) east elevation; (**b**) west elevation; (**c**) south elevation; (**d**) north elevation.

**Figure 12 materials-17-03644-f012:**
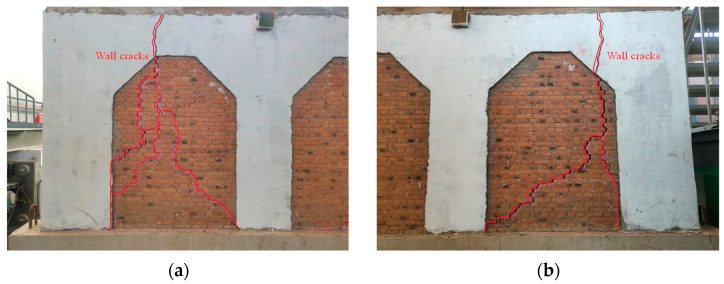
Failure mode of north elevation: (**a**) The east side; (**b**) the west side.

**Figure 13 materials-17-03644-f013:**
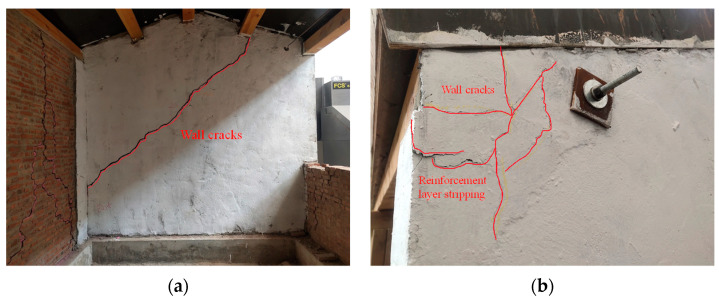
Failure mode of the east gable wall: (**a**) On the inside; (**b**) on the outside.

**Figure 14 materials-17-03644-f014:**
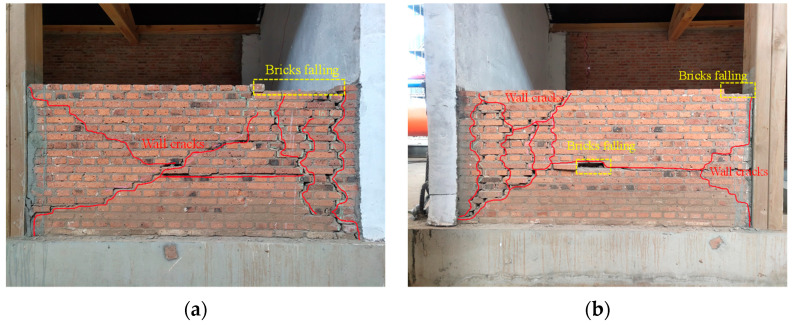
Failure mode of the cantilevered wall: (**a**) The east side; (**b**) the west side.

**Figure 15 materials-17-03644-f015:**
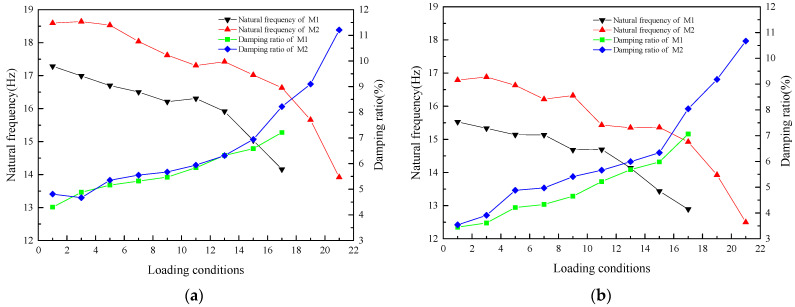
Natural frequency and damping ratio of model structure under different loading conditions: (**a**) X-direction; (**b**) Y-direction.

**Figure 16 materials-17-03644-f016:**
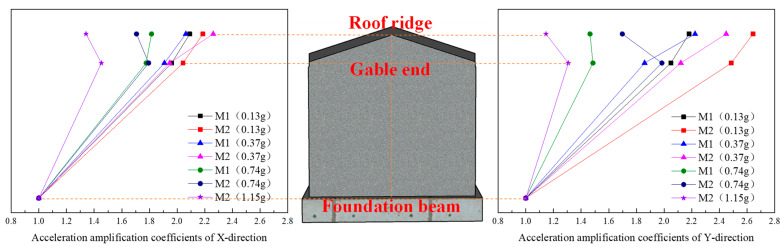
Acceleration amplification coefficients of the model under the action of El Mayor seismic waves.

**Figure 17 materials-17-03644-f017:**
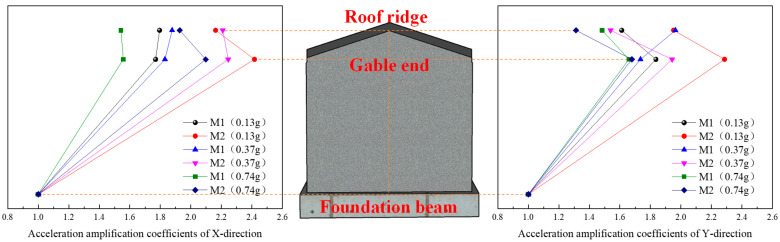
Acceleration amplification coefficients of the model under the action of Landers seismic waves.

**Figure 18 materials-17-03644-f018:**
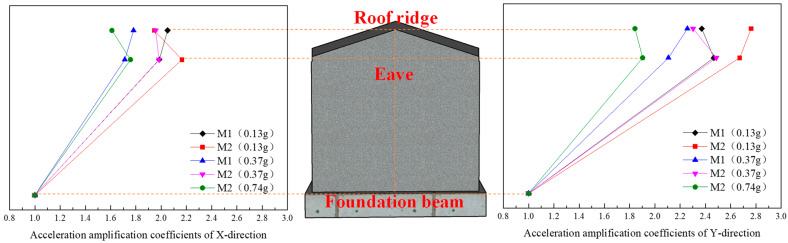
Acceleration amplification coefficients of the model under the action of artificial seismic waves.

**Table 1 materials-17-03644-t001:** Similarity coefficient of physical quantity.

Type	Physical Quantity	Similarity Coefficient
Material property	Vertical stress (S_σ_)	0.536
	Shear stress (S_τ_)	1.000
	Elastic modulus (S_E_)	1.000
	Poisson’s ratio (S_γ_)	1.000
	Density (S_ρ_)	1.000
Dynamical property	Mass (S_L_)	0.134
	Stiffness (S_X_)	0.500
	Periodicity (S_θ_)	0.518
	Frequency (S_m_)	1.929
	Acceleration (S_k_)	1.861

**Table 2 materials-17-03644-t002:** Loading conditions for shaking table test of model structure.

Loading Stage	Loading Conditions	Seismic Waves	Prototype Structure PGA/g	Shaker Table PGA/g
-	1	White noise	—	0.05
1	2	El Mayor	0.07	0.13
	3	White noise	—	0.05
	4	Landers	0.07	0.13
	5	White noise	—	0.05
	6	Artificial wave	0.07	0.13
	7	White noise	—	0.05
2	8	El Mayor	0.20	0.37
	9	White noise	—	0.05
	10	Landers	0.20	0.37
	11	White noise	—	0.05
	12	Artificial wave	0.20	0.37
	13	White noise	—	0.05
3	14	El Mayor	0.40	0.74
	15	White noise	—	0.05
	16	Landers	0.40	0.74
	17	White noise	—	0.05
	18	Artificial wave	0.40	0.74
	19	White noise	—	0.05
4	20	El Mayor	0.62	1.15
	21	White noise	—	0.05

**Table 3 materials-17-03644-t003:** Natural frequency and damping ratio of model structure.

Loading Conditions	X-Direction				Y-Direction			
	Natural Frequency of M1 (Hz)	Natural Frequency of M2 (Hz)	Damping Ratio of M1 (%)	Damping Ratio of M2 (%)	Natural Frequency of M1 (Hz)	Natural Frequency of M2 (Hz)	Damping Ratio of M1 (%)	Damping Ratio of M2 (%)
1	17.285	18.592	4.30	4.81	15.527	16.786	3.45	3.54
3	16.992	18.641	4.88	4.67	15.332	16.881	3.61	3.91
5	16.699	18.531	5.16	5.35	15.136	16.631	4.21	4.88
7	16.503	18.035	5.32	5.55	15.132	16.212	4.33	4.97
9	16.210	17.621	5.47	5.67	14.682	16.322	4.65	5.41
11	16.308	17.308	5.84	5.94	14.695	15.432	5.22	5.66
13	15.917	17.422	6.32	6.31	14.135	15.351	5.68	5.99
15	15.039	17.021	6.58	6.94	13.438	15.362	5.98	6.34
17	14.160	16.632	7.21	8.22	12.890	14.923	7.06	8.04
19	—	15.658	—	9.10	—	13.925	—	9.18
21	—	13.92	—	11.21	—	12.498	—	10.67

**Table 4 materials-17-03644-t004:** The acceleration amplification coefficients of different positions in the X-direction of the model structure.

Seismic Waves	Loading Conditions	Foundation Beam	M1		M2	
			Gable End	Roof Ridge	Gable End	Roof Ridge
El Mayor wave	2	1	1.961	2.091	2.043	2.186
	8	1	1.946	2.261	1.908	2.063
	14	1	1.794	1.706	1.776	1.815
	20	1	-	-	1.452	1.341
Landers wave	4	1	1.769	1.795	2.417	2.162
	10	1	1.830	1.877	2.245	2.209
	16	1	1.557	1.542	2.097	1.927
Artificial wave	6	1	1.985	2.050	2.164	1.945
	12	1	1.712	1.781	1.983	1.956
	18	1	-	-	1.757	1.610

**Table 5 materials-17-03644-t005:** The acceleration amplification coefficients of different positions in the Y-direction of the model structure.

Seismic Waves	Loading Conditions	Foundation Beam	M1		M2	
			Gable End	Roof Ridge	Gable End	Roof Ridge
El Mayor wave	2	1	2.049	2.180	2.486	2.643
	8	1	1.859	2.224	2.121	2.449
	14	1	1.485	1.464	1.984	1.698
	20	1	-	-	1.307	1.146
Landers wave	4	1	1.836	1.612	2.286	1.952
	10	1	1.735	1.964	1.942	1.539
	16	1	1.659	1.483	1.678	1.312
Artificial wave	6	1	2.466	2.371	2.671	2.761
	12	1	2.107	2.257	2.486	2.302
	18	1	-	-	1.903	1.842

**Table 6 materials-17-03644-t006:** Interlayer displacement and interlayer displacement angle under different loading conditions.

Seismic Waves	Direction	Loading Conditions	M1		M2	
			Displacement/mm	Displacement Angle	Displacement/mm	Displacement Angle
El Mayor wave	X	2	0.769	1/2666	0.515	1/3981
		8	2.076	1/987	1.543	1/1329
		14	11.032	1/186	3.54	1/579
		20	-	-	26.395	1/78
	Y	2	0.720	1/2847	0.586	1/3498
		8	1.184	1/1731	0.908	1/2258
		14	2.014	1/1018	1.645	1/1246
		20	-	-	10.974	1/187
Landers wave	X	4	0.868	1/2362	0.745	1/2752
		10	2.247	1/912	1.587	1/1292
		16	20.983	1/98	7.916	1/259
	Y	4	0.500	1/4100	0.421	1/4869
		10	0.818	1/2506	0.782	1/2621
		16	3.552	1/577	2.014	1/1018
Artificial wave	X	6	0.563	1/3641	0.514	1/3988
		12	2.087	1/982	1.764	1/1162
		18	-	-	8.875	1/231
	Y	6	0.683	1/3001	0.515	1/3981
		12	1.085	1/1889	0.994	1/2062
		18	-	-	2.844	1/721

## Data Availability

Data are contained within the article.

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
