# Peer review of "Seismic Performance of a Single-Story Timber-Framed Masonry Structure Strengthened with Fiber-Reinforced Cement Mortar"

_materials, 2024, doi:10.3390/ma17153644_

Round 1

Reviewer 1 Report

Comments and Suggestions for Authors

From my point of view, this text could become an interesting article. However, in my opinion, this text is not acceptable to be published yet due to the following.

1- The introduction requires improvement of several lacks. The review of previous related research in the third paragraph (lines 55-73) is too confusing. It mixes projects about testing, reinforcing with different materials…, and reaches unfounded strong statements (line 711-72). It also lacks clearly defining the gap this research project is covering, which is crucial in a scientific article. On the other hand, the introduction lacks some final lines briefly explaining the general structure of the research paper and its parts.

2- Section 2 has a similar problem because it lacks presenting and justifying in general an overview of the experimental campaign, the tests, specimens, procedures…, preferably between caption 2 and 2.1. Furthermore, there are subsections with disorganized contents, for instance 2.6, which describes from the reinforcement of the specimen to FRCM tests. This is very confusing because the two tested specimens are explained in 2.1 and 2.6, and in between there are other sections about the test program among other issues. Potential readers will get lost with this structure so it must be improved.

3- The manuscript lacks further explaining what Table 1 has symbols and abbreviations that lack their definition when first mentioned, especially for potential readers less specialized on the topic. Some similarity coefficients do not result correctly from the formulae. Moreover, lists of symbols and abbreviations would ease the understanding of future potential readers. Other concepts such as the seismic waves requires being better presented and referenced in the manuscript for non-experts readers.

4- This paper lacks a discussion of the results compared to previous related technical literature.

5- Conclusions require rewriting to solve several lacks. The first paragraph unnecessarily and too briefly tries to summarize the paper, instead it requires clearly presenting the main contribution of this paper to its field of research. The following three paragraphs present the main findings related to the results, though it is not clear.  The last paragraph presents unfounded conclusions, because this article does not compare its results to other cases in terms of economic, efficiency and mechanical performance and then it cannot state regarding to these terms.

6- The use of the word “vertical is inconsistent. It is associated to the Y plan direction (line 110) or Z direction (lines 185-186,…) and this needs to be fixed as well.

7- General final typewriting revision required to avoid minor errors. For instance, line 68 “.Kou Jialang…” no space after dot.

Reviewer 2 Report

Comments and Suggestions for Authors

This manuscript is a test report of a timber-framed masonry structure, before and after application of a Fiber Reinforced Cement Mortar (FRCM). Multiple issues were identified in this paper, such as omission of relevant and recent literature review, lack of originality, absence of methodological details, lack of discussions that derived new knowledge to the scientific field.

1.   The abstract did not state the research gap identified in the literature and selected to be investigated in the present work.

2.   The introduction section justified the development of this work within the Chinese context, focusing on the timber-framed masonry structures commonly found in rural areas of Beijing. However, publications in international journals like “Materials” should also be contextualized within a broader and global framework.

3.   The literature review is incomplete. Only a single paragraph revised previous works on the seismic performance of masonry structures.

4.   The last paragraph of Section 1 is not suitable because it appears to primarily address methodological aspects rather than fulfilling the introductory aspects expected of this section.

5.   The must clarify the research gap (issue that was never investigated in previous articles) at the end of the Introduction section.

6.   Novelty of the present study should be indicated based on the research gap identification recommended in comment #6.

7.   Section 2.1 did not present the basic characterization of bricks (strength, elastic modulus, water absorption, etc.) used to produce the masonry specimens.

8.   Section 2.1 did not present the basic characterization of wood (type, strength, elastic modulus, moisture, etc.) used to produce the structures.

9.   Raw materials, mixture proportions and production procedures used to obtain the masonry mortar were not provided.

10.                   The authors did not indicate the procedures used to construct the masonry specimens (mortar bedding approach, mortar joint thickness, workmanship quality, environmental conditions, curing period, etc.).

11.                   Section 2.3 did not justify the location selected for the sensors.

12.                   The authors did not report the age of the masonry specimens when they carried out the shaking table test.

13.                   Raw materials, mixture proportions and production procedures used to obtain the FRCM were not reported in Section 2.6.

14.                   There is no justification of the geometry of the FRCM layer presented in Figure 5.

15.                   Section 2.6 did not report the standard methods, loading rate and equipment used in the compression, flexural and bending tests of FRCM.

16.                   Section 2.6 did not report the standard methods used to cast the specimens for the compression, flexural and bending tests of FRCM.

17.                   There are two sections numbered as 3.1.1, which is not suitable.

18.                   In most subsections of Section 3, the authors mainly stated data represented in figures and tables. There should also be comparisons and synthesis of knowledge that provide new information on this field. Original contributions to the current state-of-the art were not discussed.

19.                   In Section 3, Figure 7 should show the 4 sides of the M1 specimen.

20.                   In Section 3, Figure 7 and Figure 8 should designate the sides of the masonry specimens according to the orientation nomenclature presented in Figure 2.

21.                   Originality is not highlighted in the conclusion section.

22.                   Limitations of this research were not indicated at the end of the paper.

23.                   The authors did not list recommendations for future studies in this scientific field.

24.                   High-quality vectors were not presented in some figures of this manuscript.

25.                   The scientific soundness of this paper is compromised as it resembles more of a test report on a timber-framed masonry structure rather than a comprehensive scientific article.

Comments on the Quality of English Language

Moderate editing of English language required. There are many grammar and typo mistakes in this work, such as “300mm. he width”; “The Layout and”; “follows: The cracks”; “seismic fortification intensity 8 area.”; “weste side”; etc.

Reviewer 3 Report

Comments and Suggestions for Authors

As a professor in the scientific field of engineering constructions and traffic structures, with a priority profilation for a holistic perception and evaluation of the life cycle of transport structure and buidlings , I would like to state at the outset. Reviewed manuscript "Seismic Performance of Single-Story Timber-Framed Masonry  Structure Strengthened with Fiber Reinforced Cement Mortar" I overall rate the article as excellent and extremely suitable for the journal Materials.. 

For the purpose of the possibility of immediate publication of the assessed manuscript, I would like to recommend theminor changes of a formal nature.

LNSA (line Numer of Scientific Article) 100...2.3. Instrumentation and Measurement....I recommend moving the title to the next page.

LNSA 114...Figure 3. Layout of displacement meters...it is necessary to move the figure title to its graphic part.

LNSA 120...Figure 4. Acceleration time history curves: (a) El Mayor wave; (b) Landers wave; (c) Artificial wave....when describing the y axis, it would be appropriate to indicate the meaning of the abbreviation. I recommend explaining the meaning of the physical unit gal in the  text.

LNSA 178...3.1.1. M2...subchapter has the same number as on LNSA 162. Personally, I would recommend not using the names subchapters M1, M2 and M3.

LNSA  212...Table 3. Natural frequency and damping ratio of model structure...I recommend moving the entire table to the next page.

LNSA 308...3.4. Displacement Response Analysis...I recommend moving the title to the next page.

In conclusion, I would like to heartily congratulate the authors on an excellent article. I consider the article a valuable contribution to sustainability buildings within urban engineering.

Reviewer 4 Report

Comments and Suggestions for Authors

The presented topic of the manuscript is an interesting one especially as it is tackling the topic of safety of inhabitants. There some issues the authors need to address before considering this manuscript for possible publication.

Lines 37-43 - the authors are required to remove or rephrase the paragraph. If rephrasing, please remove any reference to political parties as this is a scientific paper. Something along the line "considering the new national policy / program for increasing the safety...."

Lines 47-49 - a drawing would be highly appreciated and help the potential reader to better understand the geometry and layout of the house.

Line 90 - what do you mean by "less artificial mass model"?

Figure 2 - please indicate the X and Y axes, as stated at line 109.

Line 119 - what were the criteria of selecting the seismic records (except for the artificial one)? Where the seismic records scaled?

Figure 4 - please represent the time histories in g units (to fit with the data presented in all subsequent tables)

Line 123 - similar comment as for line 119. Please specify in more detail what El Mayor and Landers stand for (location, year and magnitude).

Section 2.6 - some photos taken during the repairing / strengthening process would highly help understanding what was done.

Figure 6b - what does KW stand for? How about 1, 2 and 3? Did you test 3 samples or 3 sets of samples?

Line 178 - Section numbering should be 3.1.2

Table 6 - how was the displacement angle actually computed?

Comments on the Quality of English Language

Please use past tense throughout the manuscript to present what has been done. Present tense should be used when referring to figures, tables or citing research paper and in the Conclusions section.

Line 52 - what do you mean by "unfortified"? Unreinforced, not strengthened?

Lines 262-264 - please consider it as a separate sentence.

Line 55 - "has always been the concern of scholars..."

Round 2

Reviewer 1 Report

Comments and Suggestions for Authors

From my point of view, this article has improved compared to the previous version but it is not acceptable to be published yet due to the following.

1- Section 2 is about the experimental program but still lacks presenting and justifying in general an overview of the experimental campaign, the tests, specimens, procedures…, preferably between caption 2 and 2.1. The most to similar to this is the last paragraph of the introduction, which is a first section that still lacks ending with a brief presentation of the paper main sections and their contents.

2- This paper still lacks a discussion of the results compared to previous related technical literature.

3- Last typewriting review required: “According to” in line 76; “The seismic performance of masonry structure” in line 53; multi- -story in line 460, …platform.And… in line 344

Comments on the Quality of English Language

Last typewriting review required: “According to” in line 76; “The seismic performance of masonry structure” in line 53; multi- -story in line 460, …platform.And… in line 344

Reviewer 2 Report

Comments and Suggestions for Authors

The paper should be rejected because the authors did not respond to the reviewer comments appropriately. Most of the responses were brief and generic, without making relevant changes to the manuscript. The reviewer response letter must clearly outline all changes made in the manuscript, following the template provided by MDPI for this purpose. For example, the following problems were identified in the revised manuscript:

1.   Response to comment #1 is not correct. The research gap added to the abstract was “seismic performance is generally poor”, which is incorrect. Extensive research already reported the seismic performance of timber framed masonry structures. Then, the abstract did not state convincing research gap identified in the literature.

2.   Response to comment #2 is not convincing. The authors did not add any sentence to contextualize the paper within a broader and global framework. Although this technology may be used in different locations, the context of the study is specific to China. Thus, claiming its applicability to single-story existing brick and wood structure houses in various countries without considering regional differences may overlook important local factors.

3.   Response to comment #3 is not satisfactory. The literature review was not complemented. In the previous version of the paper, the introduction section cited 26 works. The revised introduction section cited 27 works. The authors did not revise many recent papers on the seismic performance of masonry structures.

4.    Response to comment #4 is not satisfactory. The last paragraph of still address numerous methodological aspects rather than fulfilling the introductory aspects expected of this section.

5.   Response to comment #5 is not convincing because the authors did not add any sentence to clarify the research gap (issue that was never investigated in previous articles) at the end of the Introduction section.

6.   Response to comment #6 is not appropriate because the authors did not add any sentence to clarify the novelty of the present study, based on the research gap identification recommended in comment #5.

7.   Response to comment #7 is incomplete because Section 2.1 did not present the basic characterization of bricks (elastic modulus, water absorption, etc.) used to produce the masonry specimens.

8.   Response to comment #8 is not suitable Section 2.1 did not present the basic characterization of wood (type, strength, elastic modulus, moisture, etc.) used to produce the specimens.

9.   Response to comment #9 must be clarified. Why did you choose this mass proportion? Did you use common volume mix proportions for mortar bedding? Which one?

10.   Response to comment #10 is incomplete because the authors did not add part of the required information specimens (mortar bedding approach, workmanship quality, environmental conditions, etc).

11.   Response to comment #11 is incomplete because the authors did not explain details related to the justification for the location of the sensors, based on citation of relevant literature recommendations.

12. Response to comment #12 is questionable because the specimens were subjected to ground motion loading after 3 months of curing. In previous papers, masonry specimens are usually tested after the standard 28 days of curing.

13. Response to comment #13 is incomplete because some procedures used to obtain the FRCM were not added.

14.  The authors should also carefully respond to comments #14 to #25 of the first review round. 

Comments on the Quality of English Language

Grammar and typo mistakes were not corrected (e.g. “300mm. he width”; “was 200mm. the”; etc.). Moderate editing of English language is required.

Reviewer 4 Report

Comments and Suggestions for Authors

The manuscript is in much better shape than initially submitted and it fulfills all requirements to be accepted for publication.
